# Acute and Sub-Chronic Effects of Microplastics (3 and 10 µm) on the Human Intestinal Cells HT-29

**DOI:** 10.3390/ijerph18115833

**Published:** 2021-05-28

**Authors:** Giuseppa Visalli, Alessio Facciolà, Marianna Pruiti Ciarello, Giuseppe De Marco, Maria Maisano, Angela Di Pietro

**Affiliations:** 1Department of Biomedical and Dental Sciences and Morphofunctional Imaging, University of Messina, 98100 Messina, Italy; gvisalli@unime.it (G.V.); afacciola@unime.it (A.F.); marianna.pruiti@gmail.com (M.P.C.); 2Department of Chemical, Biological, Pharmaceutical and Environmental Sciences, University of Messina, 98100 Messina, Italy; giuseppe.demarco@unime.it (G.D.M.); maria.maisano@unime.it (M.M.)

**Keywords:** microplastics, polystyrene, cytotoxicity, ROS production, HT29 cell line, short and long term effects

## Abstract

Due to ingestion of contaminated foods, the human gastrointestinal tract is the most likely site of exposure to microplastics (MPs) with gut barrier dysfunction and intestinal inflammation. Aimed to assess the effects induced by MPs with different granulometry (polystyrene (PS) 3 and 10 µm), we performed an in vitro study by using the human intestinal cell line HT29. As a novelty, we assessed the sub-chronic exposure extending the treatment up to 48 days simulating the in vivo situation. In the range of 100–1600 particles mL^−1^, both the PS suspensions had moderate cytotoxicity after 24 h with percentages of mortality between 6.7 and 21.6 for the 10 µm and 6.1 and 29.6 for the 3 µm PS. Microscopic observation highlighted a more pronounced lysosomal membrane permeabilization in HT29 exposed to PS 3µm. Reactive oxygen species production was higher in cells exposed to PS 10 µm, but sub-chronic exposure highlighted the ability of the cells to partially neutralize this effect. Comet-assay confirmed the temporary oxidative damage that was PS-induced. Overall, considering the very fast turnover of intestinal cells, the increase in cell mortality, equal to 25% and 11% for 3 and 10 µm PS-MPs for each time point, could trigger intestinal disorders due to prolonged exposure.

## 1. Introduction

Microplastics (MPs; 1 μm to 5 mm in diameter) [1,2,3] are receiving increasing attention as persistent environmental pollutants; however, current knowledge about exposure levels and possible hazardous effects of MPs on human health are limited.

Human exposure to MPs and/or nanoplastics (NPs; ≤100 nm in diameter) primarily occurs through the oral, inhalation, dermal or parenteral routes [4]. Ingestion of MP-NP particles is likely to represent the main route of exposure in humans because they can be ingested by eating contaminated seafood [4], sugar [5], salt [6,7], beer [8], tap water [6,9], and bottled water [10,11]. Other sources of possible MPs-NPs contamination are from food packaging (e.g., plastic coffee and tea capsules) and via plants (fruits and vegetable), as they accumulate through uptake from polluted soil [12]. The presence of MPs in human stool samples (20 particles/10 g stool) confirmed MP intake in two studies [13,14], and an in vitro study by Liu et al. [15] showed that the digestive process did not alter the chemical constitution of polystyrene MPs. Based on this, the gastrointestinal tract (GIT) is the most likely site of primary exposure, although the intestinal absorption of MPs is predicted to be low [16,17].

When released into the environment, plastics are subject to slow photo- and thermo-oxidative degradation and a small amount of biodegradation, in addition to undergoing mechanical fragmentation with the formation of secondary MPs and NPs [18,19,20]. The oxygen-containing functional groups generated in the aging processes (shown by the increased carbonyl content and the presence of the hydroxyl group) modifies the physicochemical characteristics of the particle surface, making the polymers more reactive and enhancing their ability to interact with cells [21,22].

MPs can absorb several types of pollutants from the environment due to their large surface area which makes them efficient vehicles [23]. 

The quantity and type of adsorbed contaminants vary substantially, not only as a function of the residence time in the environment but also due to physicochemical changes. Furthermore, in comparison to virgin plastic particles, those modified by environmental exposure can have different adsorption affinities. Surface oxidation may increase their affinity for metals while reducing their affinity for hydrophobic compounds such as persistent organic pollutants (POPs) [24,25,26]. Taken together, this means that the overall toxicity of MPs can be extremely variable. Studies specifically designed to assess the health risk of mixtures of different vehicles and adsorbed xenobiotics are still poor because most of the uptake data on MPs have been obtained from mammalian studies that use virgin polystyrene (PS) MPs in in vivo and in vitro models [27,28,29,30,31,32]. After oral ingestion, MPs can cross the intestinal barrier and reach the systemic circulation. This translocation depends on their size and surface chemistry [33]. Furthermore, MPs can interact with a wide range of molecules, including proteins, which surround the particles forming what is known as a “corona of proteins”, significantly increasing translocation [33].

The predominant pathway for MPs uptake in the gut is through the gut-associated lymphatic tissue (GALT), specifically the microfold (M) cells of the Peyer’s patches [34]. From there, MPs pass into the bloodstream and are transported to the liver, before recirculating through the bile to the small intestine to be excreted with fecal matter [35]. Several studies involving humans, dogs, rabbits, and rodents have shown that the translocation of MPs of various types and sizes (ranging from >0.1 to 150 μm) occurs from the mammalian intestine to the lymphatic system [36]. Polyvinyl chloride (PVC) particles have been detected in the portal vein of dogs, confirming liver involvement [37]. In mice, ingested MPs could be found in the gut [38], liver [39], and kidneys [40]. 

In the gut, pathological changes induced by MPs include a reduction in mucus secretion [41], gut barrier dysfunction [42], intestinal inflammation [43], and gut microbiota dysbiosis [42,44]. In the liver, MPs induce inflammation, lipid accumulation, and lipid profile changes [41], as well as changes in markers of lipid metabolism [44]. Moreover, disorders in energy metabolism and bile acid metabolism have been observed [39,42]. 

Several studies have examined the effects of MPs on human cell cultures. Despite observing some degree of cellular uptake, in vitro studies generally show no significant signs of cellular toxicity except at very high concentrations of MPs [31,45]. Two of the most common and prominent phenomena are reactive oxygen species (ROS) production [46] and inflammatory responses [47,48]. Smaller plastic particles cross the cell membrane and induce ROS overproduction, and the resulting oxidative stress causes cytotoxicity [15,49] and DNA damage [50]. Moreover, due to MP-induced mitochondrial depolarisation, the activity of the toxicant efflux pump (ATP-binding cassette [ABC] transporter) was inhibited in Caco-2 cells [51]. 

The inflammatory effects of MPs have been confirmed by the stimulated secretion of cytokines, including IL-6, IL-8, and TNF, in monocytes, macrophages, and human peripheral blood mononuclear cells (PBMCs), and by increased histamine release from mast cell lines [52,53]. In addition, an in vitro study that investigated the effects of exposure of a co-culture of Caco-2, HT29-MTX, and Raji B to carboxylated polystyrene NPs showed changes in iron transport through the epithelial layer, resulting in the inhibition of intestinal iron absorption [54]. 

The present study aimed to assess the biological effects of exposure to polystyrene microplastics (PS-MPs) with different granulometries (3 and 10 µm in diameter) on the human intestinal epithelial cell line HT29. This is because we wanted to verify if the passage through the cell membrane and, consequently, the biological effects were size-dependent. As a novelty, we also assessed, in addition to short-term observations, the effects of subchronic exposure, extending the MP treatment up to 48 days in subsequent subcultures of the exposed cells. This was performed to simulate the in vivo effects of ingested MPs.

## 2. Materials and Methods

### 2.1. Cell Cultures and Exposure Conditions

The HT-29 epithelial cell line (ATCCR HTB-38), derived from a colorectal adenocarcinoma, was chosen because it is commonly used for the in vitro study of xenobiotic effects in the human intestinal compartment. Cells were grown in a monolayer in RPMI 1640 medium (Sigma, Milano, Italy) with 2 mM L-glutamine, 1 mM sodium pyruvate, 10% (*v*/*v*) fetal bovine serum (FBS), 100 IU/mL penicillin, and 100 μg/mL streptomycin in a humidified atmosphere containing 5% CO_2_ at 37 °C. The cells were periodically subcultured (every 4 days on average).

Experiments were performed by exposing cell monolayers to PS-MPs for the time periods and doses specified in the experimental protocol. PS-MPs with diameters of 3 and 10 µm were purchased from Sigma (Merck Life Science S.r.l., Milan, Italy) and stock suspensions were prepared in phosphate-buffered saline (PBS) (10,000 particles (p) mL^−1^). 

The choice of diameters was based on the evidence that particles of 1–4 μm of size are completely in an absorbable size range [3] while the upper limit for intracellular uptake of polystyrene plastic particles is assumed to be 10 μm [55].

To minimize PS-MPs agglomerate in liquid, due to their hydrophobicity and large surface area, the stock microplastic suspensions were sonicated for 20 min (frequency 40 kHz) prior to addition to the medium. Preliminarily microscopic observations were performed to assess if the PS-MPs were uniformly dispersed in cell medium or if, vice versa, they formed aggregates. 

In each experiment, monolayers treated with PBS instead of PS-MPs suspensions were included as negative controls (i.e., control cells). All experiments were performed in triplicate.

### 2.2. Viability Assays

The cytotoxicity induced by PS-MPs was evaluated by the colorimetric test that measures the reduction of 3-(4,5-dimethylthiazol-2yl)-2,5-diphenyltetrazolium bromide (MTT) to a formazan salt as an indicator of the activity of cellular dehydrogenases. Briefly, adherent cells in 96-well plates (1 × 10^4^/well) were treated with PS-MP suspensions ranging from 100 to 1600 p mL^−1^ in a culture medium containing 2% FBS. 

After 24 h, the medium with PS-MPs was removed, and after three washes with PBS, 0.5 mg mL^−1^ of MTT (100 μL/well) was added and the microplates were reincubated at 37 °C for 3 h. 

A mixture of 50 mM HEPES (pH 8.0) and ethanol (1:9, *v*/*v*) was added to solubilize the purple-colored formazan crystals, which were proportional to the number of live cells. Spectrophotometric measurement at 540 nm was performed using a microplate spectrophotometer reader (Tecan Italia, Milan, Italy). The values obtained were compared to the negative control, to which 100% vitality was arbitrarily assigned.

### 2.3. Short-Term Prooxidant Effect of Polystyrene Microplastics

To test the short-term prooxidant effects of PS-MPs in HT29 cells (i.e., ROS overproduction), cells were exposed to PS-MPs suspensions for 0.5, 1, 2, 3, 4, 5, 6, and 24 h. For the measurements of ROS, the highly liposoluble non-fluorescent 2’,7’-dichlorofluorescein-diacetate (DCF-DA) probe (Sigma Chemical Co., St. Louis, MO, USA) was used [56]. It is rapidly hydrolyzed by cell esterases and can rapidly oxidize in the presence of ROS, forming the highly fluorescent 2’,7’-dichlorofluorescein (DCFH).

Briefly, 80% subconfluent monolayers in 96-well plates were washed once with PBS, then 1 μM of the probe solution in 10 mM D-glucose (pH 7.4) was added to the wells and incubated at 37 °C for 30 min. After PBS washing to remove any non-internalized probe, cells were treated with PS-MPs (800 and 1600 p mL^−1^) in a cell medium containing 2% FBS, with eight wells tested for each suspension. In addition to negative controls, a positive control, 300 μM H_2_O_2_, was included in each experiment. The doses were set on the basis of the study of Hwang et al. [57], according to which it is estimated an annually maximum intake of 11,000 plastic particles per person through food, about 237,250 plastic particles by drinking water, and still on average of about 50,000 plastic particles through personal care or biomedical products. Based on these data, we estimated an intake of about 800 particles per day, opting also for a double dose to evaluate a more intense exposure.

Before the fluorimetric reading, the supernatant was recovered to evaluate the percentage of dead cells, and wells were thoroughly washed with PBS. Using a microplate fluorimeter (Tecan Italia, Milan, Italy) set to 485 and 535 nm as the excitation and emission wavelengths, respectively, the emission values were measured at time 0 and again following the time course protocol after incubation at 37 °C. The reading at time 0 was subtracted from each emission value, and ROS production was calculated as the percentage change (Δ%) compared to control cells. 

Cells suspended in the supernatant were incubated with propidium iodide (PI) solution (3 μg mL^−1^) at 4 °C for 5 min, then cell death was fluorometrically measured using 535 and 615 nm as the excitation and emission wavelengths respectively.

Short-term effects of PS-MPs were also evaluated by microscopic observation to assess the integrity of the endocytic apparatus (late endosomes and lysosomes). The metachromatic fluorophore acridine orange (AO) was used, which is captured by protons and collects in the acidic vacuolar compartment. When the probe is highly concentrated it emits red fluorescence, while green fluorescence is visible in the cytosol and nucleus where it scarcely accumulates. Decreased red fluorescence and increased green fluorescence are indicative of damage to the acid compartment. Semiconfluent HT29 monolayers grown on chamber slides (Merck Life Science S.r.l., Milan, Italy) and treated with PS-MPs suspensions (800 p mL^−1^) for 24 h were exposed to the AO solution (5 µg mL^−1^) [58]. Cells were observed by confocal laser scanning microscopy (CLSM) using TCS-SP2 (Leica Microsystems, Heidelberg, Germany) coupled to a fluorescence microscope (DM-IRE2; Leica Wetzlar, Germany) and equipped with an Ar/Kr laser.

### 2.4. Comet Assay

DNA integrity was evaluated by single-cell gel electrophoresis (i.e., Comet assay) performed according to previous studies [59]. Cells, treated as described above, were assayed in duplicate using 2 × 10^4^ cells per spot. Electrophoresis was carried out for 30 min at 300 mA and 25 V (0.86 V cm^−1^). After ethidium bromide staining (20 µg mL^−1^), 100 randomly selected nuclei were acquired using a DMIRB fluorescence microscope at 400× magnification (Leica Microsystem, Heidelberg, Mannheim, Germany) and submitted to CASP automated image analysis. Tail moment (TM), which is the product of the tail length and the fraction of total DNA in the tail, was the parameter for DNA damage.

### 2.5. Long-Term Experiments

To evaluate the effects of subchronic exposure, which is more similar to the real-life situation given the daily exposure of humans to MPs, HT29 cells were exposed to PS-MP suspensions of both types at concentrations of 800 or 1600 p mL^−1^ for 7, 14, 21, 28 and 48 days, or to PBS for the same period in control cells. 

Briefly, for these experiments, PS-MP-exposed and -unexposed monolayers were subcultured when a confluence of about 80% was achieved. An equal number of cells were subcultured for all trials, and cell progeny were reincubated at 37 °C in a growth medium (10% FSB), to which the respective PS-MPs suspension or PBS was added. The remaining cell suspensions were aliquoted to perform the analyses specified in the experimental protocol.

Cells suspended in the supernatant were incubated with PI solution (3 μg mL^−1^) at 4 °C for 5 min, and mortality was fluorometrically measured using 535 and 615 nm as the excitation and emission wavelengths, respectively.

### 2.6. Statistical Analysis

Statistical analyses were performed by using GraphPad Software for Science (San Diego, CA, USA). The relationship between the different variables was assessed by calculating the Pearson correlation coefficient, and significance was accepted at *p* < 0.05.

## 3. Results

To assess the acute effects of exposure to 3 and 10 µm PS-MPs, we initially evaluated cytotoxicity in our cell model using the MTT test. The assayed dose ranged from 100–1600 p mL^−^^1^. Up to the concentration of 160.000 p mL^−1^ for 3 and 10 µm PS-MPs respectively, the preliminary microscopic observation did not show any aggregation of both particles in the suspensions in the cell medium (Appendix A). 

Both PS-MP suspensions had a moderate cytotoxic effect, reducing the rate of cell viability, which was positively related to the exposure dose (Table 1). The absorbance values at the lowest exposure dose were only about 6% lower than the control cells for both types of MPs. As the exposure dose increased, the effect differed according to particle size, with the percentage of dead cells ranging between 6.7% and 21.6% for 10 µm PS-MPs and between 6.1% and 29.6% for 3 µm PS-MPs. The significant correlation between dose and effect was confirmed by Pearson’s test, with r values of 0.98 and 0.82 for 10 and 3 µm PS-MPs, respectively (*p* < 0.01). 

Microscopic observation of the endocytic apparatus confirmed the viability values. As shown in Figure 1, more pronounced lysosomal membrane permeabilization was observed in HT29 cells exposed to 3 µm PS-MPs, which showed diffuse green cytosolic fluorescence, indicating damage to late endosomes and lysosomes (Figure 1B). On the other hand, cells exposed to the larger PS-MPs showed a lysosomal compartment more similar to control cells (Figure 1A,C).

### 3.1. Short-Term Reactive Oxygen Species Production Induced by Polystyrene Microplastics

Figure 2 reports the time course of ROS levels in cells exposed to 800 and 1600 p mL^−1^ PS-MP suspensions, expressed as the percentage change (Δ%) compared to control cells. Similarly, for the cell viability assessment, ROS production by PS-MP-exposed cells had moderate effects, especially when compared to that observed in the positive control (i.e., cells treated with 300 µM H_2_O_2_).

The elevated ROS production induced by PS-MP exposure was very rapid, already evident at 0.5 h. Over time, ROS levels progressively increased, with the peak level recorded at 6 h and a slight decrease at 24 h, when further increases were recorded in the positive control.

No correlation was observed between dose and effect. At the highest doses for both PS-MPs, the emission values were only 5% higher than those recorded for the lower dose. Instead, a clear and significant correlation between exposure time and effect was observed, and the Pearson r coefficients were always >0.90 (*p* < 0.01), similar to that observed for the positive control. In comparison to the 3 µm PS-MPs, larger plastic particles caused a faster and slightly greater increase in ROS production, with fluorescent emission values approximately double at 0.5 h, on average, and at least 25% higher for the other time points examined.

### 3.2. Long-Term Effects of Exposure to Polystyrene Microplastics 

To evaluate the effects of subchronic exposure, simulating what happens in individuals who constantly ingest water or other beverages and food contaminated by MPs, the progeny of PS-MP-exposed intestinal cells, also subjected to the same treatment, were periodically monitored for up to 48 days. As reported above, 3 and 10 μm PS-MP suspensions that had been freshly prepared in growth medium at concentrations of 800 and 1600 p mL^−1^ were used at each step to set up the subcultures (every week on average).

With regard to ROS production, although PS-MP exposure caused increased ROS production in the short term, this increase was no longer observable in the progeny of exposed cells after at least 7, 14, and 21 days. As shown in Figure 3A, in the 7-day experiment, the average emission values were comparable to control cells. After 14 days, a further decrease was observed, with average ROS production reduced by 21.82% and 36.20% in cells exposed to 10 and 3 µm PS-MPs, respectively, compared to control cells. Although there were smaller decreases, a similar trend was observed even after 21 days of exposure. By prolonging the treatment, the ability of cells to neutralize the pro-oxidant effect of PS-MPs was reduced, and the fluorochrome emission values were higher than control cells after 28 and 48 days of exposure. In fact, the values were, on average, higher for both sizes of microplastics at 28 days and especially at 48 days, with an average increase of about 18.36%. Even in the long-term experiments, ROS production was not related to exposure dose, especially for the 3 µm PS-MPs. 

In long-term experiments, the fluorimetric analysis of dead cells, performed in parallel to the ROS analysis, highlighted limited effects throughout the exposure period. At the highest exposure dose (1600 p mL^−1^) of both microplastics, the recorded values were superimposable to the controls; however, increased cell mortality was recorded in cells treated with the 800 p mL^−1^ dose. As this parameter was calculated as the ratio of PI emission values in the treated monolayers to the control, the increase in cell mortality over the entire period was, on average, equal to 25% and 11% for 3 and 10 µm PS-MPs, respectively.

Similar to the findings for ROS production, the results of the comet assay show highly fragmented DNA following short-term treatment, with an increase in TM compared to control cells of 29.5 and 66.8 Δ% for 3 and 10 µm PS-MPs (1600 p mL^−1^), respectively. However, the effects of long-term exposure in terms of DNA damage were not evident after up to 28 days of exposure, as TM values were higher than control cells only after 48 days of exposure to 3 and 10 µm PS-MPs, with increases of 4.5 and 11.5, respectively (Figure 3B). Significant r values, based on the Spearman test, were obtained between TM values and ROS levels (*p* = 0.0028).

## 4. Discussion

To improve our understanding of the potential adverse health effects of ingested MPs in humans, we used an in vitro model resembling subchronic exposure of the intestinal epithelium. This approach allowed us to verify the toxicity both in the short-term and after prolonged exposure to these unavoidable ubiquitous xenobiotics, previously found to induce acute cytotoxicity in other intestinal cell lines [31,45,50,51,57]. 

Similar to all experimental studies conducted to date, virgin PS-MPs were used in our study. This represents a limitation because they do not correspond to MPs present in natural environments, however, regardless of this limitation, our results highlight the importance of the size of MPs. Although the same number of particles were present, the higher cytotoxicity elicited by smaller MPs can undoubtedly be attributed to easier internalization by cells not specialized in phagocytosis, such as epithelial HT29. This was confirmed by damage to the endocytic apparatus by lysosomal membrane permeabilization, which was exclusively observed in cells exposed to 3 µm PS-MPs. Particles of at least 60 nm in size can be internalized by energy-dependent endocytosis machinery, through vesicles derived from the plasma membrane and involving actin polymerization by the stepwise use of GTPases [60,61]. In addition to being unique to specialized cells (phagocytosis in professional phagocytes), this mechanism is common in almost all cells (including intestinal epithelial cells), which can internalize micro- and nanoparticles by pinocytosis. In both endocytosis mechanisms, particle-cell surface interactions, favored by the hydrophobicity of PS-MPs, induce membrane invagination and the formation of vesicles containing foreign particles, which are internalized and subsequently fuse with lysosomes to form phagosomes with a diameter of 0.5–10 μm. Our results highlight the damage of phagosomes in cells exposed to 3 µm PS-MPs, producing an effect known as the “lysosome-enhanced Trojan horse effect”, characterized by intracellular toxicity and apoptosis due to acidification, as well as the enzymolysis of lysosomes [62]. 

Contrary to our observations regarding PS-MP-induced acute cytotoxicity, which was dose-dependent and higher in cells exposed to 3 µm PS-MPs, the production of ROS was slightly higher and not dose-dependent in cells exposed to the 10 µm PS-MPs. Although the latter does not seem to cause lysosomal membrane permeabilization following internalization, the slightly higher ROS production suggests oxidative DNA damage in these cells, as confirmed by the Comet assay.

The absence of a dose-response correlation of ROS production is not an abnormal result as other studies using PS-MPs while showing significant differences compared to the control cells, highlighted no significant differences with respect to concentration [57,63,64]. It is plausible to believe that the absence of a dose-effect response is attributed to the rapidity with which the cells, depending on the stimulus undergone, counteract the pro-oxidant effect.

It is likely that the larger particles follow a different pathway. Indeed, in general, ROS production in cells is known to occur in two ways: ROS production from the Mitochondrial Electron Transport Chain (ETC) during routine aerobic respiration, or production of ROS through the oxidative bursts of NADPH oxidases (NOXs) [65]. The former could be attributed to mitochondrial impairment, while the latter is normally a consequence of bacterial invasion, as NOXs are activated by bacterial products and cytokines. However, this innate immunity mechanism is also triggered by xenobiotics [66] leading to what is known as “sterile inflammation”, i.e., an inflammatory response without pathogenic infection [67]. MPs have been demonstrated to act as stressors, either by causing mitochondrial depolarisation [51] or through an inflammatory process [57]. 

The different behaviors of the two PS-MPs in terms of the acid compartment and ROS production are consistent with the results of Wu et al. [51], who found that 0.1 μm PS-MPs accumulated in lysosomes but larger particles (5 μm) did not, as observed by confocal fluorescence microscopy imaging. The latter could escape from the phagosomes and randomly localize in the cell cytoplasm, causing ROS production [51] and subsequent oxidative DNA damage. Our Comet assay data confirm that PS-MPs can induce the breakage of DNA strands, albeit transiently, probably due to oxidative stress [68].

The results of subchronic exposure suggest good adaptability of the examined biological model, as the observed effects were transient. This highlights the remarkable homeostasis capabilities of HT29, as underlined by the ROS values in the examined intermediate time points, which were even lower than the control cells, and confirmed by the results of the Comet assay. 

Although extending the exposure time seemed to reduce the ability of cells to balance the damage, ROS overproduction induced by PS-MPs was very weak after long-term exposure. However, the biological effects of virgin PS-MPs do not seem to be without long-term consequences. Intestinal epithelial cells have a fast turnover, and our experiments highlighted a moderate increase in mortality of cell progeny after prolonged exposure in comparison to control cells. This suggests that, independently from the ability to adsorb polluting compounds during their stay in the environment and from their increased reactivity due to abiotic degradation, PS-MPs could trigger intestinal disorders. 

## 5. Conclusions 

Overall, the results obtained in our study show moderate acute effects and, to a lesser extent, subchronic effects of PS-MP exposure. Even if the differences in sizes of the examined PS-MPs were relatively small, the uptake occurred by different pathways, causing damage to the acid compartment and a higher increase of ROS in 3 and 10 µm particles respectively. This could lead to diversified effects as the remarkable ability of the cells to counteract the weak pro-oxidant effect, would neutralize, or in any case limit, the damage induced by the 10 µm PS-MPs. On the contrary, the greater impacts of PS-MPs in the cell lines HT29 would seem to consist of damage to the acid compartment which is notoriously irreversible and incompatible with the maintenance of cell viability. On these bases, the exposure to virgin plastic particles would not seem devoid of consequences to human cells. Further investigations are necessary to assess the health risks related to exposure to these ubiquitous persistent pollutants, which will inevitably be ingested with water, beverages, and several foods.

## Figures and Tables

**Figure 1 ijerph-18-05833-f001:**
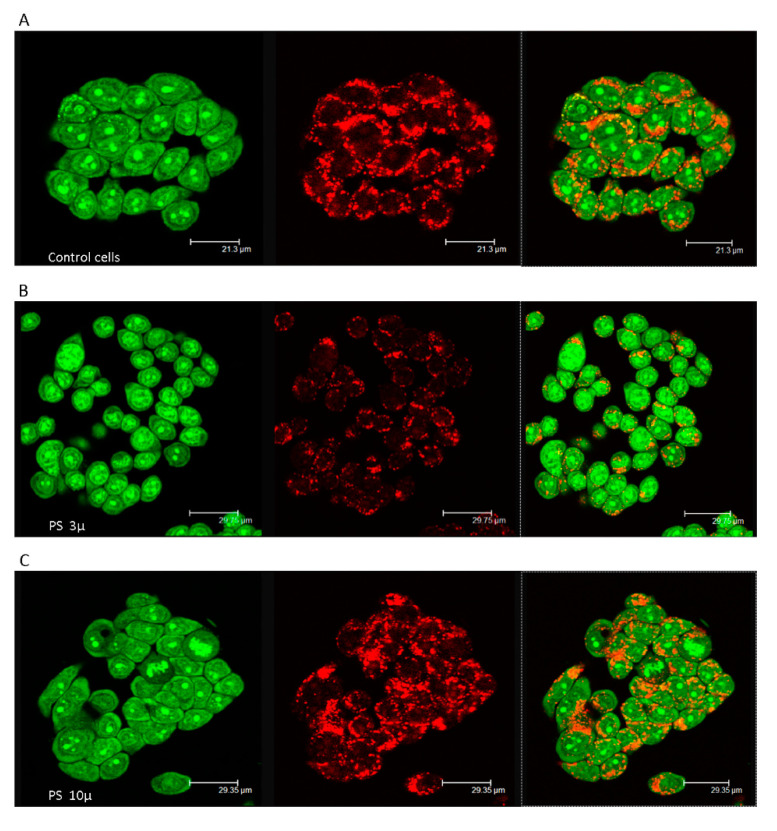
Membrane destabilization of the acidic vacuolar compartment in PS- MPs exposed cells. Images of acridine orange (AO)-loaded cells (400x magnifications) are reported: (**A**) Untreated cells showing intact lysosomes. (**B**,**C**) HT-29 treated with PS- MPs (800 p mL^−1^ for 24 h) 3 and 10 µ respectively. Diffuse green cytosolic fluorescence and the reduced number of red dots, in B images highlight the damage to late endosomes and lysosomes.

**Figure 2 ijerph-18-05833-f002:**
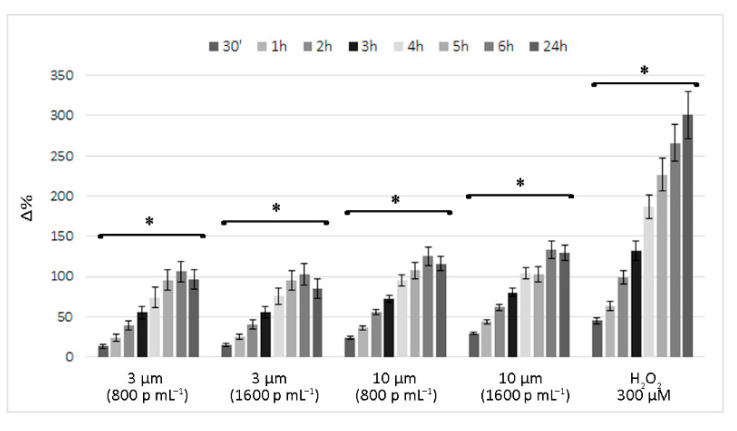
Time course of reactive oxygen species (ROS) production in PS- MPs exposed cells and in the positive control (H_2_O_2_ treated cell). The values are expressed as Δ% in comparison to untreated cells. * Similarly to the positive control, a significant correlation between exposure time and ROS production was observed (* *p* < 0.05) for both MPs and at the two doses tested.

**Figure 3 ijerph-18-05833-f003:**
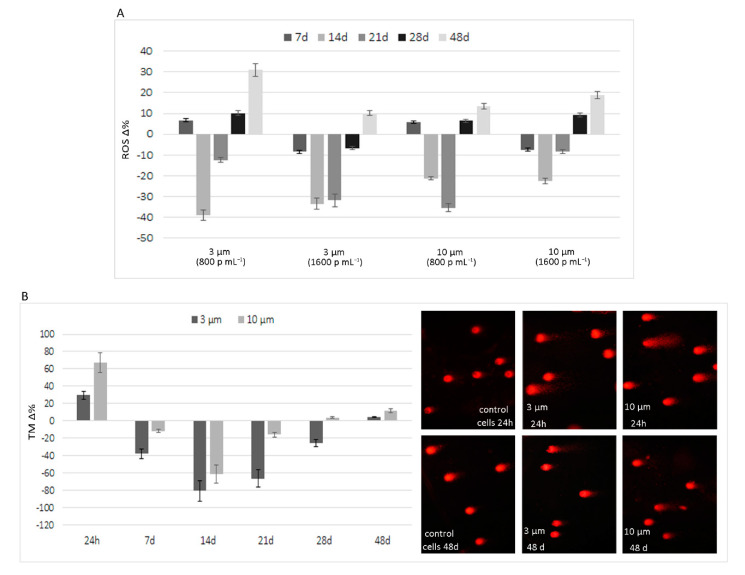
Effects of sub-chronic exposure in PS- MPs treated cells. (**A**) Δ % of ROS production in comparison to untreated cells after 7, 14, 21, 28, and 48 days. The graph highlights the ability of HT-29 to neutralize, albeit temporarily, the pro-oxidant effect of PS-MPs. (**B**). Results of Comet assay expressed as Δ % of tail moment ™ in comparison to untreated cells. Short-term treatment (24 h) caused a highly fragmented DNA in cells exposed to 3 and 10 µm PS-MPs (1600 p mL^−1^) which was suppressed in the intermediate time points and then only partially reappeared after prolonged exposure, showing the same trend of ROS production. On the right are shown representative images of Comet assay in control and PS-MPs treated cells after 24 h and 48 days.

**Table 1 ijerph-18-05833-t001:** Results of the MTT cell viability test after exposure to polystyrene (PS)-MPs (microplastics) for 24 h. Values are expressed in % of dead cells compared to control cells, in bracket standard deviation values.

PS	100 p mL^−1^	200 p mL^−1^	400 p mL^−1^	800 p mL^−1^	1600 p mL^−1^
10 µ	6.70	6.31	8.78	15.01	21.55
	(0.71)	(0.76)	(0.81)	(1.34)	(2.15)
3 µ	6.05	15.99	18.68	21.37	29.63
	(0.68)	(1.21)	(1.61)	(1.95)	(3.01)

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
