# Peer review of "Acute and Sub-Chronic Effects of Microplastics (3 and 10 µm) on the Human Intestinal Cells HT-29"

_ijerph, 2021, doi:10.3390/ijerph18115833_

Round 1
Reviewer 1 Report
Abstract: Abbreviations need to be introduced fireplace they are using (PS, ROS, etc.).
Introduction:
Lines 1: Microplastics definition: The given definition according to the cited reference is understandable. However, the commonly accepted definition is 1 micron to 5 mm. The cited reference is 2016; the discussion on microplastics has been evolved lot since 2016. Hence the definitions also need to be up to the date. Suggest using a better and up to date citation on this case. (Unless this definition has a strong support to describe the continent of the paper, if there is a huge relevance, then better justify you are following this classification of microplastics and Nano-plastics).
Methods:
Abbreviation PBS (phosphate buffer saline) shall be introduced full form at the first place. There are several other similar cases (i.e. MTT).
Lines 123-What is the reason for using this concntration of microplastics? “cells were 122 treated with PS-MPs (800 and 1600 p mL-1)”.? Authers shall mention the reasons and justify.
Results:
Table 1: Units (it says p/mL and microgram/mL). Which is correct?
Discussion: The first three paragraphs seems tobe more suitable for intorduction. Specially, discussion about environmetal pollutents which are potentailly adsorbed into MPs etc. Discussion shall be based on your results and related literature.
Lies 310-330: Authers discuss about larger and smaller paticles, but what is more releent her is the relative size of the particles used in the study. Cosidering the large range of sizes of MPs, the 3 and 10 both seems to be same range. If not authors shall mention it clearly.
Author Response
AUTHOR'S REPLY TO THE REVIEW REPORT (REVIEWER 1)
Abstract: Abbreviations need to be introduced fireplace they are using (PS, ROS, etc.).
R: As suggested, we added the full denomination of abbreviations at the first place.
Introduction:
Lines 1: Microplastics definition: The given definition according to the cited reference is understandable. However, the commonly accepted definition is 1 micron to 5 mm. The cited reference is 2016; the discussion on microplastics has been evolved lot since 2016. Hence the definitions also need to be up to the date. Suggest using a better and up to date citation on this case. (Unless this definition has a strong support to describe the continent of the paper, if there is a huge relevance, then better justify you are following this classification of microplastics and Nano-plastics).
R: Although the definition of microplastics as plastic particles with a diameter of less than 5 mm is quite dated, even today most researchers adopt 5 mm as the upper size limit for microplastics and we added the following references:
- Sorensen RM, Jovanović B. From nanoplastic to microplastic: A bibliometric analysis on the presence of plastic particles in the environment. Mar Pollut Bull. 2021 Feb;163:111926. doi: 10.1016/j.marpolbul.2020.111926. Epub 2020 Dec 18.
- Stock V, Fahrenson C, Thuenemann A, Dönmez MH, Voss L, Böhmert L, Braeuning A, Lampen A, Sieg H. Impact of artificial digestion on the sizes and shapes of microplastic particles. Food Chem Toxicol. 2020 Jan;135:111010.
Methods:
Abbreviation PBS (phosphate buffer saline) shall be introduced full form at the first place. There are several other similar cases (i.e. MTT).
R: We added the full form of the abbreviations.
Lines 123-What is the reason for using this concntration of microplastics? “cells were 122 treated with PS-MPs (800 and 1600 p mL-1)”.? Authers shall mention the reasons and justify.
R: In the new version, we added the following sentences (page 4, lines 154-160):
“The doses were set on the basis of the study of Hwang et al. (2020) according to which it is estimated an annually maximum intake of 11,000 plastic particles per person through food, about 237,250 plastic particles by drinking water, and still on average of about 50,000 plastic particles through personal care or biomedical products. Based on these data, we estimated an intake of about 800 particles per day, opting also for a double dose to evaluate a more intense exposure”.
Results:
Table 1: Units (it says p/mL and microgram/mL). Which is correct?
R: We changed microgram/mL with p/mL and we apologize for the mistake.
Discussion: The first three paragraphs seems tobe more suitable for intorduction. Specially, discussion about environmetal pollutents which are potentailly adsorbed into MPs etc. Discussion shall be based on your results and related literature.
R: We have transferred some parts of the discussions into the introduction (page 2 lines 44-59), as suggested.
Lies 310-330: Authers discuss about larger and smaller paticles, but what is more releent her is the relative size of the particles used in the study. Cosidering the large range of sizes of MPs, the 3 and 10 both seems to be same range. If not authors shall mention it clearly.
R: We changed smaller with 3 µm and larger with 10 µm. Although the differences in MP size may seem minor, the study shows that biological effects are diversified as a function of particle size. Moreover, we added the sentence (page 3, lines 117-119): “The choice of diameters was based on the evidence that particles of 1–4 μm of size are completely in an absorbable size range (Stock et al., 2020) while the upper limit for intracellular uptake of polystyrene plastic particles is assumed to be 10 μm (Bruinink et al., 2015).
Reviewer 2 Report
L212: Might there be an explanation for why there is a slight decrease between 6h and 24h across all studies whereas there is a continued increase in the positive control?
L222-223 and L259-260: Will these data be put in an appendix / supplementary information? If not, these seem like unnecessary / unsubstantiated sentences.
L242-243: ROS production being unrelated to exposure dose doesn’t necessarily appear to be true if the experiments only considered two levels of exposures. What were the motivations for picking these two levels of exposures? Do we know if the levels of exposure chosen reflect any kind of meaningful value…in terms of minimum / maximum / average ingested amounts of PS-MPs in a given diet or something? It could also be that the doses are too high such that dose-dependent trends are obscured, especially if we expect the trends to reach an asymptote, for example.
L248 / Figure 3B: If a positive delta percentage tail moment indicates damage to DNA, what does a negative delta percentage tail moment mean?
Author Response
AUTHOR'S REPLY TO THE REVIEW REPORT (REVIEWER 2)
L212: Might there be an explanation for why there is a slight decrease between 6h and 24h across all studies whereas there is a continued increase in the positive control?
R: We believe that cells are able to activate defence systems, which manage to neutralize the limited pro-oxidant effect of polystyrene and this can explain the differences compared to positive control. Conversely, the exposure to hydrogen peroxide causes a strong time-dependent damage that cells are unable to counteract. The remarkable homeostasis capabilities of the examined cell model were also observed after a sub-chronic exposure to PS-MPs and a very weak ROS overproduction was observed exclusively in the longest exposure times.
L222-223 and L259-260: Will these data be put in an appendix/supplementary information? If not, these seem like unnecessary/unsubstantiated sentences.
R: We eliminated the sentences as suggested.
L242-243: ROS production being unrelated to exposure dose doesn’t necessarily appear to be true if the experiments only considered two levels of exposures. What were the motivations for picking these two levels of exposures? Do we know if the levels of exposure chosen reflect any kind of meaningful value…in terms of minimum / maximum / average ingested amounts of PS-MPs in a given diet or something? It could also be that the doses are too high such that dose-dependent trends are obscured, especially if we expect the trends to reach an asymptote, for example.
R: We have chosen the doses 800 and 1600 p/mL because, on the basis of the viability test, they represent the highest concentrations that ensure about 70% of cell viability necessary for the study concerning the damage PS-NPs-induced in sub-chronic exposure experiments. Moreover, we added the sentence (page 3, lines 117-119): “The choice of diameters was based on the evidence that particles of 1–4 μm of size are completely in an absorbable size range (Stock et al., 2020) while the upper limit for intracellular uptake of polystyrene plastic particles is assumed to be 10 μm (Bruinink et al., 2015).
Surely, the use of only two doses could represent a limit in the detection of significant differences. However, the absence of a dose-response correlation is not an abnormal result. Many other studies using PS-MPs, while showing significant differences compared to the control cells, showed no significant differences with respect to size or concentration. The following references were added in the new version:
- Hwang, J.; Choi, D.; Han, S.; Jung, S.Y.; Choi, J.; Hong, J. Potential toxicity of polystyrene microplastic particles. Sci. Rep. 2020, 10(1), 7391, doi: 10.1038/s41598-020-64464-9.
- Wang, Q.; Bai, J.; Ning, B.; Fan, L.; Sun, T.; Fang, Y.; Wu, J.; Li, S.; Duan, C.; Zhang, Y.; et al. Effects of bisphenol A and nanoscale and microscale polystyrene plastic exposure on particle uptake and toxicity in human Caco-2 cells. Chemosphere 2020, 254 ,126788, doi: 10.1016/j.chemosphere.2020.126788.
- Rubio, L.; Barguilla, I.; Domenech, J.; Marcos, R.; Hernández, A. Biological effects, including oxidative stress and genotoxic damage, of polystyrene nanoparticles in different human hematopoietic cell lines. J. Hazard. Mater. 2020, 398, 122900, doi: 10.1016/j.jhazmat.2020.122900.
L248 / Figure 3B: If a positive delta percentage tail moment indicates damage to DNA, what does a negative delta percentage tail moment mean?
R: As already reported in the results, the effects of a long-term exposure in terms of DNA damage were higher than control cells only after 48 days of exposure to 3 and 10 µm PS-MPs. At the shortest exposure times, TM value of the exposed cells was similar and in some cases lower compared to the control cells, confirming the ability of the examined cell model to neutralize the weak pro-oxidant effect of PS-MPs.
Reviewer 3 Report
This manuscript titled “Acute and sub-chronic effects of microplastics (3 and 10 μm) on 2 the human intestinal cells HT-29” aimed to assess the biological effects of exposure to polystyrene microplastics (PS-MPs) with different granulometries on the 80 human intestinal epithelial cell line HT29, especially the effects of subchronic exposure. The subject addressed in this article was worthy of investigation. This manuscript was well organized and could provide some new information about the toxic effect of microplastics on human. However, there were still some issues needed be answered or revised before publication.
- Why 3 and 10 μm in diameter were chosen? Is the particle size difference between the two too small? Especially considering the aggregation effect of PS-MPs.
- Why 100, 200, 400, 800, 1,600 μg mL-1 were chosen?
- p120, how many times washed with PBS? How do you non-internalised probe were totally removed? Why not use serum free medium (SFM) to wash? PBS saves money, but it does not keep stem cells alive for long, which will affect the subchronic exposure experiment.
- What was the particle size of PS-MP in the suspensions? Did the particle size change because of the aggregation?
- p211, “ROS levels progressively increased, with the peak level recorded at 6 h and a slight decrease at 24 h, when further increases were recorded in the positive control”, why?
- p 213, “No correlation was observed between dose and effect”, why?
- The cell viability and ROS production were affected by many factors over time. How do you ensure that the results were not influenced by other factors?
Author Response
AUTHOR'S REPLY TO THE REVIEW REPORT (REVIEWER 3)
- Why 3 and 10 μm in diameter were chosen? Is the particle size difference between the two too small? Especially considering the aggregation effect of PS-MPs.
R: Even if the differences in size are relatively small, our data shows that the uptake is markedly different, occurring through different pathways. These are responsible for a greater damage to the acid compartment for 3 µm PS-MPs while the 5 µm ones are relatively more pro-oxidant. Moreover, in the new version we added the following two sentences with the related references:
“The choice of diameters was based on the evidence that particles of 1–4 μm of size are completely in an absorbable size range [Stock, 2020] while the upper limit for intracellular uptake of polystyrene plastic particles is assumed to be 10 μm [Bruinink, 2015].
- Stock, V.; Fahrenson, C.; Thuenemann, A.; Dönmez, M.H.; Voss L.; Böhmert, L.; Braeuning, A.; Lampen, A.; Sieg, H. Impact of artificial digestion on the sizes and shapes of microplastic particles. Food Chem. Toxicol. 2020, 135, 111010, doi: 10.1016/j.fct.2019.111010;
- Bruinink, A.; Wang, J.; Wick, P. Effect of particle agglomeration in nanotoxicology. Arch. Toxicol. 2015, 89(5), 659-75, doi: 10.1007/s00204-015-1460-6.
- Why 100, 200, 400, 800, 1,600 μg mL-1 were chosen?
R: After consulting the available literature, we chose a rather high range to have more accurate indications on biological effects PS-MPs-induced.
Based on reported data, it is estimated an annually maximum intake of 11,000 plastic particles per person through food, about 237,250 plastic particles by drinking water, and still on average of about 50,000 plastic particles through personal care or biomedical products. Based on these ranges, we estimated an intake of about 800 particles per day, opting also for a double dose to evaluate a more intense exposure.
- Hwang, J.; Choi, D.; Han, S.; Jung, S.Y.; Choi, J.; Hong, J. Potential toxicity of polystyrene microplastic particles. Sci. Rep. 2020, 10(1), 7391, doi: 10.1038/s41598-020-64464-9.
- p120, how many times washed with PBS? How do you non-internalised probe were totally removed? Why not use serum free medium (SFM) to wash? PBS saves money, but it does not keep stem cells alive for long, which will affect the subchronic exposure experiment.
R: We added how many times we washed with PBS and how we removed the non-internalized probe. The washes with PBS are rather rapid, therefore they do not compromise the cell viability.
- What was the particle size of PS-MP in the suspensions? Did the particle size change because of the aggregation?
R: As reported in the supplementary material, microscopic observations showed that in PS-MPs suspensions in the cell medium used in our experiment, the plastic particles were uniformly dispersed and not aggregated up to 48h, allowing a good interaction with the cells.
- p211, “ROS levels progressively increased, with the peak level recorded at 6 h and a slight decrease at 24 h, when further increases were recorded in the positive control”, why?
R: We believe that cells are able to activate defence systems that neutralize the longer-term effects of polystyrene. In fact, as commented in the discussions, it is highlighted a good homeostatic capability of HT29, as the observed effects were transient. Exposure to hydrogen peroxide, on the other hand, causes time-dependent damage that the cells are unable to counteract.
- p 213, “No correlation was observed between dose and effect”, why?
R: While cell viability was positively related to the exposure dose, ROS production did not show a dose-response effect. It is plausible to believe that the absence of a dose-effect response is attributable to the rapidity with which the cells, depending on the stimulus undergone, counteract the pro-oxidant effect. The absence of a dose-response correlation is not an abnormal result. Many other studies using PS-MPs, while showing significant differences compared to the control cells, showed no significant differences with respect to size or concentration. The following references were added in the new version:
- Hwang, J.; Choi, D.; Han, S.; Jung, S.Y.; Choi, J.; Hong, J. Potential toxicity of polystyrene microplastic particles. Sci. Rep. 2020, 10(1), 7391, doi: 10.1038/s41598-020-64464-9.
- Wang, Q.; Bai, J.; Ning, B.; Fan, L.; Sun, T.; Fang, Y.; Wu, J.; Li, S.; Duan, C.; Zhang, Y.; et al. Effects of bisphenol A and nanoscale and microscale polystyrene plastic exposure on particle uptake and toxicity in human Caco-2 cells. Chemosphere 2020, 254 ,126788, doi: 10.1016/j.chemosphere.2020.126788.
- Rubio, L.; Barguilla, I.; Domenech, J.; Marcos, R.; Hernández, A. Biological effects, including oxidative stress and genotoxic damage, of polystyrene nanoparticles in different human hematopoietic cell lines. J. Hazard. Mater. 2020, 398, 122900, doi: 10.1016/j.jhazmat.2020.122900.
- The cell viability and ROS production were affected by many factors over time. How do you ensure that the results were not influenced by other factors?
R: The presence of other confounding factors may represent a limitation of the study. However, the presence of control cells kept under the same conditions as the exposed cells should ensure that the differences found are attributable to exposure to microplastics.
Round 2
Reviewer 1 Report
(REVIEWER 1): Lines 1: Microplastics definition:
The authors have not addressed the point raised by the reviewer. What R1 has pointed was, the definition is 1 micron to 5 mm. Also, the added references [2 and 3] is not updated version, it is recitation of 2016 document.
Please address this (my suggestion is to change it to 1 micron to 5 mm).
Lines 491-493: the value 160.000 seems to be a mistake (or can not recognize since the figure in supplementary material has not been labeled or described it properly).
Also, authors need to follow the journal specifications to prepare the supplementary material. (need proper numbering of figures etc. also need to mention it properly in the manuscript. Please look carefully at the instructions for the author.
Figure 2 caption: "significant corre-550 lation between exposure time and effect...", I can not find the relevence of * mark (also I can not see any * mark in the figure). Please make this clear to the readers.
Lines 607-609: The 30% and 20% are not excat values. but authors hace mentioned in a misleading way (see the Figure 3A, the values are different from 20% and 30%). Pelase correct this. (also correct similar kind of other cases. "After 14 days, a fur-607 ther decrease was observed, with average ROS production reduced by 20% and 30% in 608 cells exposed to 10 and 3 μm PS-MPs, respectively, compared to control cells." ---- Recomend to write exact values.
Minor comment: please try to keep the uniformity (some cases authors write 1,600 and some cases it is written as 1600).
Author Response
Comments and Suggestions for Authors
(REVIEWER 1):
Lines 1: Microplastics definition:
The authors have not addressed the point raised by the reviewer. What R1 has pointed was, the definition is 1 micron to 5 mm. Also, the added references [2 and 3] is not updated version, it is recitation of 2016 document.
Please address this (my suggestion is to change it to 1 micron to 5 mm).
R: we apologize for misinterpreting the reviewer's comment, we have changed the diameter range of the microplastics as suggested (line 29)
Lines 491-493: the value 160.000 seems to be a mistake (or can not recognize since the figure in supplementary material has not been labeled or described it properly).
Also, authors need to follow the journal specifications to prepare the supplementary material. (need proper numbering of figures etc. also need to mention it properly in the manuscript. Please look carefully at the instructions for the author.
R: Sorry for the mistake, we have renamed the figure and mentioned in the text according to the guidelines for additional data (line 215), adding a more accurate description as required.
Figure 2 caption: "significant correlation between exposure time and effect...", I can not find the relevence of * mark (also I can not see any * mark in the figure). Please make this clear to the readers.
R: We have already added the mark "*" in the figure and resubmitted, but we note that the previous version of figure 2 is present in the text. We resubmit the correct figure 2 again (line 245).
Lines 607-609: The 30% and 20% are not excat values. but authors hace mentioned in a misleading way (see the Figure 3A, the values are different from 20% and 30%). Pelase correct this. (also correct similar kind of other cases. "After 14 days, a further decrease was observed, with average ROS production reduced by 20% and 30% in cells exposed to 10 and 3 μm PS-MPs, respectively, compared to control cells." ---- Recomend to write exact values.
R: we wrote the exact values (lines 272 and 279).
Minor comment: please try to keep the uniformity (some cases authors write 1,600 and some cases it is written as 1600).
R: we changed 1600 with 1,600
Reviewer 3 Report
The manuscript has been improved by the authors based on the comments of the reviewer. The manuscript in the current form can be accepted for publication.
Author Response
We thank the reviewer for his positive comment.